# Learning Robust Penetration Testing Policies under Partial Observability: A systematic evaluation

**Raphael Simon**                                                                 *r.simon@cylab.be*
*Cyber Defence Lab, CISS Department*
*Royal Military Academy*
*AI Lab, Department of Computer Science*
*Vrije Universiteit Brussel*

**Pieter Libin**[*]
*AI Lab, Department of Computer Science*
*Vrije Universiteit Brussel*

**Wim Mees**[*]
*Cyber Defence Lab, CISS Department*
*Royal Military Academy*

**Reviewed on OpenReview:** *https://openreview.net/forum?id=YkUV7wfk19*

## Abstract

Penetration testing, the simulation of cyberattacks to identify security vulnerabilities, presents a sequential decision-making problem well-suited for reinforcement learning (RL) automation. Like many applications of RL to real-world problems, partial observability presents a major challenge, as it invalidates the Markov property present in Markov Decision Processes (MDPs). Partially Observable MDPs require history aggregation or belief state estimation to learn successful policies. We investigate stochastic, partially observable penetration testing scenarios over host networks of varying size, aiming to better reflect real-world complexity through more challenging and representative benchmarks. This approach leads to the development of more robust and transferable policies, which are crucial for ensuring reliable performance across diverse and unpredictable real-world environments. Using vanilla Proximal Policy Optimization (PPO) as a baseline, we compare a selection of PPO-based variants designed to mitigate partial observability, including frame-stacking, augmenting observations with historical information, and employing LSTM or TrXL architectures. We conduct a systematic empirical analysis of these algorithms across different host network sizes. We find that this task greatly benefits from history aggregation. Converging up to four times faster than other approaches. Manual inspection of the learned policies by the algorithms reveals clear distinctions and provides insights that go beyond quantitative results.

## 1 Introduction

The world is more interconnected than ever. Dependence on this connectivity has become deeply ingrained in our society. Never before have so many computer systems been deployed to operate our critical infrastructure, such as health, finance, transportation and energy. At the same time, cyber-related threats continue to grow, posing significant risks, especially for these critical infrastructures (Macaulay, 2019; World Economic Forum, 2023). Defending those systems is a significant challenge, especially given the fundamental asymmetry; defenders have to cover every possible security flaw while attackers often only need a single point of entry. One way to strengthen the security posture of companies and organizations is to perform penetration testing

---

[*]Equal contribution, alphabetical order.

and red teaming, in which ethical hackers try to uncover vulnerabilities or implement specific techniques and procedures to emulate advanced adversaries, such as nation-state-sponsored actors (Libicki et al., 2015). This allows defenders to uncover blind spots in their detection capabilities and be better prepared for future attacks. It also enables them to practice and refine their responses in case of a real attack. Conducting these security assessments requires experienced professionals, who are in short supply (ISC$^2$, 2023), and the process is time-consuming and costly. In their work, penetration testers use numerous tools the community has developed, which allow for some automation of the work. However, automation of the overall process remains an unresolved challenge, yet is crucial to help professionals scale their efforts to secure more systems and protect critical infrastructure.

The key difficulty lies in decision-making: skilled practitioners must continuously evaluate partial information, adapt to dynamic network conditions, and choose from a vast array of possible actions. These sequential decision-making challenges, involving long-term consequences, are well-suited to formulation as reinforcement learning (RL) problems. RL's ability to learn optimal behaviour through trial and error makes it particularly suited for penetration testing. Just as human penetration testers (or pentesters) learn to improve their strategies through experience, RL agents learn policies that balance immediate rewards with long-term outcomes, optimizing cumulative performance over time (Sutton & Barto, 2018). This parallel becomes particularly compelling with the emergence of Deep Reinforcement Learning (DRL), which has achieved superhuman performance in complex strategic environments, from mastering Atari games (Mnih et al., 2015) to pioneering self-play approaches in competitive domains like Dota (Berner et al., 2019) and Go (Silver et al., 2017). Further successes in complex real-world applications include the design of a magnetic controller for nuclear fusion in a tokamak configuration (Degrave et al., 2022) to learning prevention strategies in the context of pandemic influenza (Libin et al., 2021). Inspired by these achievements, researchers have started applying DRL to automate the penetration testing process.

Several environments have been proposed in the literature to enable both the simulation of penetration testing and facilitating the training of RL agents (Schwartz & Kurniawatti, 2019; Standen et al., 2021; Microsoft Defender Research Team, 2021; Janisch et al., 2023; Oesch et al., 2024). Learning directly in real-world environments remains an outstanding challenge, not only due to the sample inefficiency of RL algorithms (Dulac-Arnold et al., 2021), but also the time cost of resetting virtualized infrastructure for each episode. Regarding the training of agents in simulated environments, we find that most of the work formalises the problem of penetration testing as a Markov Decision Process, allowing the agent to perceive the full state of the environment (Zhou et al., 2021; Li et al., 2023; Tran et al., 2022). While this assumption may be reasonable in certain scenarios, it has long been criticized as an oversimplification of real-world conditions, for penetration testing, where information gathering is a core component of the task (Sarraute et al., 2013). Framing the problem as partially observable requires the agent to seek information, evaluate it, and then act accordingly, mimicking human penetration testers. Another challenge current methods are facing is overfitting (Zhang et al., 2018; Cobbe et al., 2019), where agents learn the training environment by heart and as such do not generalise to previously unseen scenarios. In the context of penetration testing, learning policies that are applicable to multiple network configurations is a crucial step toward achieving robustness and real-world applicability.

To address these challenges, this work formalizes the network penetration testing task as a partially observable and stochastic decision-making problem. Since existing environments do not highlight these challenges sufficiently, we make the necessary adaptations to the Network Attack Simulator (NASim) and provide a new environment we call StochNASim. First, we add stochasticity related to the network topology by generating a new permutation of the network every episode. The hosts will be re-generated and have new properties such as which processes, services and operating system (OS) they are running. Second, we enlarge the observation space to account for different network sizes. As a result, the agent will have to act successfully in an environment composed of 5 hosts in one episode, while in the next episode it might encounter a network of 8 hosts. This increases the number of possible observations the agent perceives, rendering the environment more challenging. It also highlights the challenge to learn a policy that effectively generalizes. Using this environment, we evaluate distinct techniques for addressing partial observability that have shown promise in prior DRL applications. Our selection includes frame stacking and observation augmentation, as well as more expressive architectures such as recurrent neural networks and Transformer-XL (TrXL). All methods

are PPO-based, and compared against vanilla PPO as a baseline, which has no specific mechanism for handling partial observability. To ensure a fair comparison, we conduct a comprehensive hyperparameter search for each algorithm and verify these hyperparameter sets on several control seeds to assure their robustness. Through this systematic evaluation, we seek to better understand effective strategies for addressing partial observability and stochastic dynamics in the context of automated penetration testing.

We summarize our contributions as follows:

1. We adapt NASim to create StochNASim, a new partially observable and stochastic penetration testing environment with variable network sizes that better reflects real-world challenges and allows the learning of robust and transferable policies.

2. We conduct a systematic empirical evaluation of different PPO-based approaches for handling partial observability in penetration testing scenarios.

3. We demonstrate that simple observation augmentation can significantly outperform complex architectures (LSTM, TrXL), challenging conventional assumptions about memory mechanisms in this domain.

4. We provide comprehensive policy analysis through action sequence visualization, revealing qualitative differences between algorithmically similar solutions that go beyond quantitative performance metrics.

## 2 Background

### 2.1 Penetration Testing

Penetration testing (pentesting) is a systematic cybersecurity methodology where security professionals simulate adversarial attacks to identify and exploit vulnerabilities before malicious actors can leverage them (National Institute of Standards and Technology, 2008). The typical pentesting workflow consists of five sequential phases: (1) *reconnaissance* to discover hosts and services; (2) *vulnerability scanning* to identify exploitable weaknesses; (3) *exploitation* to gain initial access; (4) *privilege escalation* to reach critical assets; and (5) *documentation* of findings and remediation recommendations. Testing methodologies vary by prior knowledge: *white-box testing* provides complete system information; *black-box testing* simulates external attackers with no prior knowledge; and *gray-box testing* offers partial knowledge to simulate insider threats (National Institute of Standards and Technology, 2022). Critically, pentesters operate with incomplete information—network topologies are discovered incrementally, host configurations remain unknown until scanned, and exploitation success depends on unpredictable factors. This sequential decision-making under partial observability makes penetration testing well-suited for reinforcement learning (Sarraute et al., 2013). We formalize penetration testing as a sequential decision-making problem where: **states** represent network configurations (hosts, services, access levels); **actions** correspond to pentesting operations (scanning, exploitation, privilege escalation); **observations** reflect partial information from each action; **rewards** incentivize progress while penalizing costly actions; and **transitions** capture action outcomes. We now introduce the formal frameworks that enable learning automated pentesting policies from this formulation.

### 2.2 Markov Decision Processes

Markov Decision Processes (MDPs) are a class of stochastic sequential decision processes in which the cost and transition functions depend only on the current state of the system and the current action (Puterman, 1990). An MDP is formalized as a tuple $\langle \mathcal{S}, \mathcal{A}, \mathcal{P}, \mathcal{R}, s_I, \gamma \rangle$ where $\mathcal{S}$ is the set of all *states* the environment can take upon; $\mathcal{A}$ a set of *actions* that can be executed in the environment; $\mathcal{P}(s'|s, a)$, the transition probability distribution over next states, conditioned on the current state and action; $\mathcal{R}(s, a) \rightarrow \mathbb{R}$, a reward function; $s_I \in \mathcal{S}$, the initial state, which can itself be a distribution; and $\gamma \in [0, 1]$ a discount factor, that signifies the importance of future rewards. MDPs are characterized by the Markov property: given the current state, the future is independent of the past. The objective in an MDP is to learn a policy $\pi(a|s)$, a probability

distribution over actions conditioned on the current state, that maximizes the expected cumulative discounted reward $G_t = \mathbb{E}_\pi[\sum_{t=0}^{\infty} \gamma^t r_t]$, which we call the return, and where $r_t$ is a random variable that represents the reward obtained at time step $t$. In finite-horizon tasks the sum is clipped to $T$, the maximum number of allowed steps.

### 2.3 Partially Observable Environments

In many real-world scenarios, the true state of the environment cannot be observed, necessitating the framework of Partially Observable Markov Decision Processes (POMDPs). A POMDP extends the MDP framework to a tuple $\langle \mathcal{S}, \mathcal{A}, \mathcal{P}, \mathcal{R}, \Omega, \mathcal{O}, s_I, \gamma \rangle$, where $\Omega$ represents the observation space and $\mathcal{O}(o|s', a)$ defines the observation function. In POMDPs, the Markov property holds for the hidden states, but not for the observations, which provide only partial information about the environment. Therefore, the agent must maintain a belief state $b_t(s)$, representing a probability distribution over possible states at time $t$. The optimal policy must now map beliefs to actions: $\pi(a|b)$.

### 2.4 Reinforcement Learning

Reinforcement Learning (RL) provides a general framework for learning optimal behavior in MDPs and POMDPs through interaction with an environment (Sutton & Barto, 2018). An RL agent learns from experience rather than requiring a complete model of the environment. At every time step $t$, the agent in state $s_t \in \mathcal{S}$ picks an action $a_t \in \mathcal{A}$. It then receives a reward $r_{t+1}$ for the outcome of that action, and perceives the new state $s_{t+1}$. This interaction loop is depicted in Figure 1.

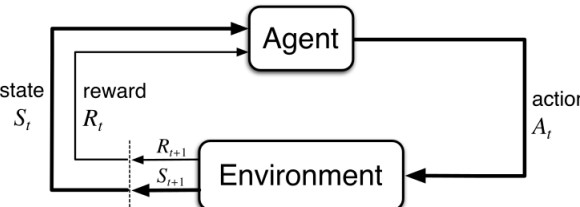

Figure 1: The agent–environment interaction in reinforcement learning. Adapted from (Sutton & Barto, 2018).

For MDPs, common RL approaches include value-based methods (e.g., $Q$-learning), policy gradient methods, and hybrid actor-critic algorithms, each with distinct trade-offs in sample efficiency, computational complexity, and suitability for discrete or continuous action spaces. Value functions estimate how good it is for an agent to be in a given state. The value function is defined as: $V_\pi(s) \doteq \mathbb{E}[G_t|s_t = s]$. It tells us the value of the state the agent is in at time step $t$ when following policy $\pi$ from there onwards. Another commonly used value function is the $Q$-function, or the state-action value function: $Q_\pi(s,a) \doteq \mathbb{E}[G_t|s_t = s, a_t = a]$. It defines the value of being in state $s$ at time step $t$, taking action $a$, and following the policy $\pi$ from there onwards.

For POMDPs, RL approaches typically incorporate mechanisms to handle partial observability. These include belief state computation (Kaelbling et al., 1998) and recurrent neural networks (RNNs) to implicitly encode the history (Hausknecht & Stone, 2017). Recent advances have explored attention mechanisms and transformer architectures to better capture long-term dependencies in observation history (Parisotto et al., 2020; Pleines et al., 2025).

## 3 Related Work

Several simulation environments have been developed to enable RL research in penetration testing, each with different design choices regarding scope and complexity. NASim (Schwartz & Kurniawatti, 2019) provides configurable penetration testing scenarios through configuration files and supports both full and partial observability modes. CybORG (Standen et al., 2021) was designed as an AI Gym for autonomous cyber

operations, supporting both *offensive agents* (which attack systems) and *defensive agents* (which detect and respond to attacks) while offering partial observability options. CyberBattleSim (Microsoft Defender Research Team, 2021) focuses specifically on *lateral movement* (moving between already-compromised hosts) and *privilege escalation* (elevating access on a single host) rather than the full penetration testing workflow that includes initial reconnaissance and vulnerability discovery, using a node-based simulation with detailed network configurations. More recent environments have built upon these foundations. NASimEmu (Janisch et al., 2023) extends NASim with an emulation component to enable evaluation in more realistic settings, while Cyberwheel (Oesch et al., 2024) supports both red and blue team operations with actions mapped to the MITRE ATT&CK framework (Strom et al., 2018).

While these environments support partial observability, most subsequent research has used fully observable MDP formulations (Zhou et al., 2021; Li et al., 2023; Tran et al., 2022; Li et al., 2024a; Yang & Liu, 2022). In contrast, we focus on POMDPs, where agents must actively discover network properties through reconnaissance: a fundamental aspect of real-world penetration testing.

Recognizing this gap, several recent works have begun to address partial observability in automated penetration testing. The dominant approach has been to augment RL algorithms with recurrent architectures. Zhang et al. (2022) combine Double DQN with LSTM, while Li et al. propose EPPTA, replacing PPO's feed-forward networks with RNNs. Ren et al. (2024) follow the same principle, also adding an LSTM architecture to PPO, but additionally incorporate an intrinsic curiosity module to help with exploration. Beyond RNN approaches, Li et al. (2024b) incorporate reward machines (Icarte et al., 2022) to encode domain knowledge by providing intermediate rewards when the agent transitions between predefined states. Terranova et al. (2024) initially formulate penetration testing as a POMDP but acknowledge that standard DRL algorithms struggle with high partial observability. To address this, they augment the observation space with historical information (e.g., previously exploited vulnerabilities), effectively transforming the POMDP into an augmented MDP.

While these works represent important progress, they have several limitations that motivate our systematic study. They typically evaluate on fixed network topologies, limiting assessment of policy generalization and risking overfitting to specific configurations. Moreover, they compare against limited baselines without systematically exploring methods for handling partial observability across varying network settings. In particular, they do not contrast computationally expensive RNN-based architectures with simpler alternatives such as frame stacking or observation augmentation, leaving it unclear whether architectural complexity is necessary. Finally, the lack of rigorous hyperparameter tuning raises questions about whether reported performance gains reflect true algorithmic improvements or better-tuned baselines.

## 4 Methodology

First, we present a formalisation of our environment. Next, we discuss the different algorithms that have been selected to address the presented issues. Finally, we provide a detailed overview of our hyperparameter search procedure.

### 4.1 Environment

In this work, we extend the Network Attack Simulator (NASim) (Schwartz & Kurniawatti, 2019). NASim simulates a network of hosts organized into subnets—logical subdivisions that group connected devices based on network requirements. Firewalls are positioned between subnets to control traffic flow, either allowing or blocking communication in specific directions. In this environment, the goal is to obtain root privileges on two hosts marked as *sensitive*. This is analogous to real-world exercises, where some hosts are more important than others. Reasons for classifying some hosts as sensitive include: containing sensitive data, or serving as major gateways into other network segments.

We chose to extend this simulator because it provides a good balance of realism and computational efficiency, offers a simple yet flexible framework, and captures the key challenges of penetration testing. What we found lacking was the ability to train on permutations of the same network configuration, which is required for testing algorithms' capabilities for learning in stochastic environments and evaluating how well policies

Table 1: Comparison of NASim and StochNASim environments

| Feature | NASim | StochNASim |
|---|---|---|
| Network topology | Fixed per scenario | Regenerated each episode |
| Network size | Fixed (e.g., 5 or 8 hosts) | Variable (e.g., 5-8 hosts) |
| Initial state | Single fixed state | Distribution of initial states |
| Host properties | Static across episodes (OS, services, processes) | Regenerated each reset (OS, services, processes) |
| Observation space | Fixed size $(m_c + 1) \times n$ where $m_c =$ current hosts | Variable size $(m + 1) \times n$ where $m =$ max hosts |
| Action space | Fixed per scenario | Regenerated each reset (padded with No-Op) |
| Stochasticity | Action success probability only | Action success probability + network generation |

generalize. To address this limitation, we extended the environment to support networks of variable size to add additional complexity. We now formalise the environment, which we call StochNASim[1], and describe all the important components and modifications we introduced to investigate algorithms for learning policies in partially observable and stochastic networks of variable size.

### 4.1.1 State

The overall network state is defined by the collective states of all individual hosts within it. A *host vector* encodes all the information about a particular host. The shape of the network state is a matrix of shape $m_c \times n$, where $m_c$ is the current number of hosts in the generated network and $n$ is the length of the host vector. The features of a host, as encoded in the host vector, are: The subnet and host address, flags indicating whether the host has been compromised, is reachable, discovered, and sensitive. Continuing, the vector contains the discovery value of the host and the attacker's current access level on it. Finally, which OS, services, and processes it is running. An example network with host properties in showcased in Figure 2. When an agent interacts with the environment, only four host values can change: *compromised*, *reachable*, *discovered*, and *access level*. All other values remain static after the environment is generated. The agent's perceived state includes the state matrix, plus four action outcome flags: action success, connection error, permission error, and undefined error. This additional information is padded with zeros to match the host vector length and added as a new row to the state matrix. As a result, the information fed to the agent is of the shape $(m_c + 1) \times n$.

### 4.1.2 Initial State

The initial state determines the position of every host within the network, including whether they are compromised, reachable, discovered, sensitive, and their current access level. It also encodes the OS, services, and processes running on each host. When the network is generated, every host is assigned a subset of running services and processes. Specifically, from the set of all available services $\mathbb{S}$ and processes $\mathbb{P}$, each host runs $n_s$ services and $n_p$ processes, where $n_s < |\mathbb{S}|$ and $n_p < |\mathbb{P}|$. Assigning only a subset of the total available services and processes to hosts is to make exploitation more difficult, requiring the agent to carefully investigate the host before selecting the action. Due to using a variable number of hosts and generating them anew, StochNASim has a distribution of initial states instead of a stationary initial state like in NASim. The agent starts on one host at the edge of the network. We note that the agent remains stationary in the network topology, acquiring deeper access to hosts rather than navigating between them.

---

[1]Code and the StochNASim environment are available at https://github.com/raphsimon/StochNASim.

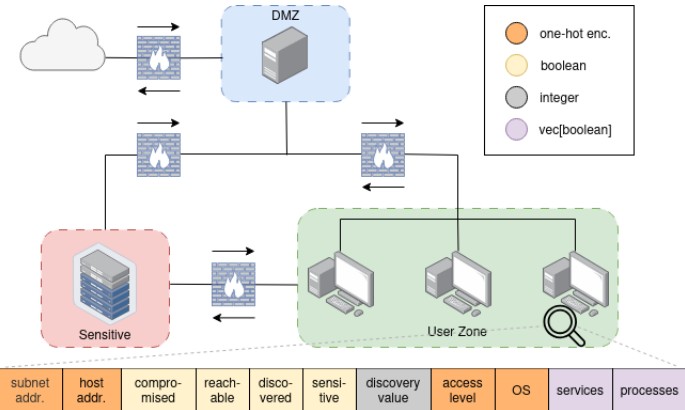

Figure 2: Example network topology in StochNASim showing five hosts across different security zones. The diagram illustrates key host properties including subnet membership (DMZ, User Zone, Sensitive), operating systems, running services, and access levels.

### 4.1.3 Actions

The environment defines seven different action types: two exploitations, four scans, and one `No Op` (do nothing) action. The two exploitation types are: `Exploit`, which exploits a vulnerable service (i.e., perform remote exploitation) to obtain initial access on a host. Each exploit action targets a specific service and OS combination. To successfully exploit a host, it needs to run the targeted service and OS. `Privilege Escalation`: exploits a vulnerable process (i.e., perform local exploitation) to elevate one's privileges on a host. Privilege escalation actions need the host to run a specific OS and process pair to run successfully. The scan actions are: `Service Scan`, `Process Scan` and `OS Scan`, which scan a host for its services, processes, and OS respectively. Finally, the `Subnet Scan` action scans the subnet the host belongs to, which allows to uncover further parts of the network. Each action is described by the host it targets, its cost, success probability and the required access level.

### 4.1.4 Action Space

The overall action space scales with the number of hosts. For each host, there is a set of scan actions $\mathbf{S} = \{\texttt{Service Scan}, \texttt{Process Scan}, \texttt{OS Scan}, \texttt{Subnet Scan}\}$. To maintain one consistent set of actions that works against every host, we associate one `Exploit` action for every service and OS combination, and one `Privilege Escalation` action for every process and OS. Therefore, the size of the action space is denoted as $|\mathcal{A}| = \lceil H \rceil \times (|\mathbf{S}| + |OS| \times (|S| + |P|))$ where $H$ is the number of hosts, $S$ is the number of services per host, and $P$ the number of processes per host configured in the environment.

### 4.1.5 Observations

In the partially observable case we consider, observations are action-specific and transient: they contain only the direct result of the action just executed. For example, if the agent scans host $h_3$ for its OS at timestep $t$, the observation at $t$ reveals $h_3$'s OS. However, at timestep $t + 1$, if the agent scans a different host or attribute, the previous information ($h_3$'s OS) is no longer present in the observation. This transient nature creates a memoryless observation stream where past discoveries are immediately forgotten unless explicitly retained. Observations encode discovered attributes as positive values (1 for binary features like 'runs SSH'), while undiscovered or unscanned attributes are represented as zeros. Importantly, when a scan reveals an attribute is absent (e.g., a service scan shows a host does *not* run HTTP), that feature also appears as zero in the observation, indistinguishable from features that have never been scanned. This reflects the realistic constraint that penetration testing tools report what they find, with negative results (absent attributes) being implicit. This observation model reflects real penetration testing workflows where tools like Nmap (Lyon, 2009) return only the specific information requested (e.g., open ports from a port scan), and where maintaining comprehensive notes across multiple reconnaissance actions is essential for success. To

allow for variable-sized networks, we use observation sizes of $(m + 1) \times n$, where $m$ is the maximum number of hosts considered and $m_c \leq m$.

### 4.1.6   Transition Function

To transition from one state to another, the agent needs to execute actions. Scan actions are deterministic, while `Exploit` and `Privilege Escalation` actions have a success probability $p$ assigned to them, following a Bernoulli distribution. State transitions occur when actions succeed based on this probability and when all necessary preconditions are met. These preconditions are:

- Each action requires that the target host is reachable.

- The `Process Scan` requires a *user* access level to succeed, while the `Subnet Scan` requires *root* privileges.

- When performing an `Exploit` or `Privilege Escalation` action, the target host must run the respective service or process that is targeted by that action.

- Additionally, `Privilege Escalation` requires the *user* access level on the host. This implies that first, a successful `Exploit` is required to gain the necessary access.

- Once an action has been successfully executed, repeating it has no further effect on the state.

### 4.1.7   Network Generation

NASim's network generation procedure creates penetration testing scenarios with a structured topology that mimics real-world enterprise networks. The generator allocates hosts across multiple subnets following a specific formula: for every 40 hosts in the network, one is assigned to the DMZ subnet, one to the Sensitive subnet, and the remainder are distributed among User subnets (with 5 hosts per subnet at most). This creates a hierarchical structure with an Internet subnet (1 host), DMZ, Sensitive, and User subnets arranged in a binary tree topology. Each host is configured with an OS, a subset of available services, and a subset of available processes. To ensure the network presents a viable penetration testing challenge, the generator guarantees exploitable paths to sensitive hosts through three mechanisms: ensuring each subnet contains at least one vulnerable host, making all sensitive hosts vulnerable to root-level access, and configuring firewall rules to allow at least one vulnerable service between network zones. The generator creates exploit actions for each OS-service combination and privilege escalation actions for each OS-process combination, ensuring that attackers can chain together exploits to traverse the network topology and ultimately compromise the sensitive hosts.

### 4.1.8   Rewards

The reward function is defined by Equation 1. The value of hosts is denoted by $V_h$. By scanning the subnet, new hosts can be discovered. The value of discovering a host is denoted by $V_d$. The reward is multiplied by the number of *newly* discovered hosts, $n_{dh}$. Hosts may only be discovered once. In all other scenarios, the returned reward is the cost of the action $a_t$.

$$r_t = \begin{cases} -cost(a_t) + V_d \times n_{dh} & \text{for successful} \\ & \text{subnet scans,} \\ -cost(a_t) + V_h & \text{for successful} \\ & \text{priv. esc.,} \\ -cost(a_t) & \text{else.} \end{cases} \tag{1}$$

### 4.1.9   Stochasticity

We argue that introducing additional stochasticity is crucial in the domain of penetration testing, as it enables learning more robust policies applicable to a diverse range of networks with varying services and processes,

rather than policies that overfit to a single network configuration. By default, the only stochasticity in the original environment is the success probability of actions. To enable learning in such a challenging scenario, we have extended NASim in several key areas, creating what we call StochNASim. Our modifications include: First, we extended the observation space to accommodate a maximum of $m$ hosts, representing the upper-bound of our variable network scenarios. Second, we generate a new network upon each environment reset, effectively sampling a new initial state $s_I \subset \mathcal{S}$. While the number of current hosts $m_c$ remains within defined bounds, the specific services and processes running on each host are regenerated with each reset. Third, we regenerate the action space with each reset to align with the initial state, to ensure a set of valid actions. While the target host address (subnet, host number) changes to match the current network configuration, all other action properties remain consistent—reflecting the stable toolbox used by real security professionals. We define one exploit action for every OS-service combination and one privilege escalation action for every OS-process combination. When the current number of hosts ($m_c$) is less than the maximum ($m$), we pad the remaining action space with `No Op` actions and return a connection error in the observation should such an action be invoked.

These modifications result in an environment that supports networks of varying sizes up to $m$ hosts, where each reset generates a new network with hosts having different properties. The consistency in action types ensures that the agent learns to associate action $a_i$ with its function rather than with specific network configurations, promoting generalization across different network topologies.

## 4.2 Algorithm Selection

We now present the selection of different algorithms and methods to address the partial observability of the StochNASim environment. Acting in POMDPs is a challenging task, and finding efficient approaches to solve them is still an active area of research (Ghosh et al., 2021; Avalos et al., 2024). We focus on model-free methods that have shown success in prior research and have been applied to tasks similar to ours. Our algorithm selection encompasses both established approaches from the penetration testing literature and promising techniques from the broader POMDP research community.

### 4.2.1 Baseline

As a baseline, we consider PPO, a policy gradient method that directly optimizes the policy parameters. The policy gradient theorem provides the theoretical foundation for algorithms like REINFORCE, which update the policy parameters in the direction of the gradient $\nabla_\theta J(\theta)$, where $J(\theta) = \mathbb{E}_{\tau \sim \pi_\theta}[G_t]$ (Williams, 1992). Here, $\tau$ denotes a trajectory, a sequence of states, actions and rewards. In other words, the objective is to maximize the expected return of trajectories sampled from the current policy. However, vanilla policy gradient methods suffer from high variance and poor sample efficiency.

Actor-critic algorithms address these limitations by incorporating a value function to reduce variance. Instead of using Monte Carlo returns, such methods use an advantage estimate $A(s,a) = Q(s,a) - V(s)$, combining the benefits of both value-based and policy-based learning. Trust Region Policy Optimization (TRPO) further improves upon basic actor-critic methods by constraining policy updates to prevent catastrophically large changes that could destabilize learning (Schulman et al., 2015).

PPO builds on TRPO's insights while offering a simpler, more practical implementation. Rather than explicitly constraining the trust region through second-order optimization, PPO uses a clipped surrogate objective that effectively limits policy. This clipping mechanism prevents excessively large policy updates while maintaining computational efficiency. PPO's robustness across domains, stable training characteristics, and widespread implementation availability make it an appropriate baseline for our comparative study of partial observability approaches (Schulman et al., 2017), as well as already having been used extensively in this domain (Terranova et al., 2024; Li et al.; Ren et al., 2024; Janisch et al., 2023).

### 4.2.2 Frame-stacking

This technique was first used in the vanilla DQN paper (Mnih et al., 2015) to provide temporal context through observation history. Frames refer to the game frames of Atari. Frame-stacking provides a small

short-term temporal context by combining the last $f_n$ observations with the current observation into a single input. Prior work has shown that frame-stacking works well on several partially observable tasks, and is able to achieve comparable performance to recurrent architectures (Cobbe et al., 2020). So far, this method has not been applied to automated penetration testing. To enable frame-stacking, we employ the frame stack wrapper found within the sb3 library (Raffin et al., 2021). We use the last $f_n$ frames, whereby $f_n$ is a hyperparameter we tune: $f_n \in \{4, 8, 16, 32\}$. Going forward, we abbreviate PPO with frame-stacking as PPO-FS.

### 4.2.3 Augmented Observations

In partially observable environments, agents must retain discovered information throughout episodes to build complete state representations. The dynamics of the environment and the structure of the state allow us to retain the information that has been obtained throughout the episode, mirroring how professional penetration testers maintain detailed notes of discovered vulnerabilities, services, and system configurations throughout their assessment. In the case of RL, this lets us acquire a representation that converges to the state of the environment as the agent explores and uncovers more information during the episode. We do this by implementing a wrapper around our environment that stacks an aggregated matrix of observations below the latest observation. The aggregation matrix is obtained by applying an element-wise maximum $O_t^{aug} = \max(O_{t-1}^{aug}, O_t)$ between itself and the latest observation, $O_t$. At timestep $t$, the top part of the observation contains the latest observation and the bottom part contains the aggregation of all the observations up to and including timestep $t-1$. We depict a simplified visualisation in Equation 2. Through this we obtain an explicit representation of the history. For instance, if a host's OS is discovered at timestep 5, this information remains visible in all subsequent observations through the aggregated matrix. The reason we stack the latest observation and the augmented matrix together, and not just use the aggregation matrix as the observation, is to provide a better signal to the agent. If we only provide the aggregated history, we end up with actions that map to the same observation, which complicates the learning process. We also omit the additional information about action outcomes from the aggregated matrix to only track state information. We call these augmented observations, and use them in conjunction with PPO. Going forward, we abbreviate PPO with augmented observations as PPO-AO.

$$O_1^{aug} = \begin{bmatrix} 0 & 1 & 0 \\ 0 & 0 & 1 \\ 0 & 0 & 0 \\ 0 & 0 & 0 \end{bmatrix}, \quad O_2^{aug} = \begin{bmatrix} 1 & 0 & 0 \\ 0 & 0 & 1 \\ 0 & 1 & 0 \\ 0 & 0 & 1 \end{bmatrix}, \quad O_3^{aug} = \begin{bmatrix} 0 & 0 & 1 \\ 1 & 0 & 0 \\ 1 & 1 & 0 \\ 0 & 0 & 1 \end{bmatrix} \tag{2}$$

### 4.2.4 Recurrent Architectures

To handle the sequential nature of observations in our partially observable environment, we evaluate PPO combined with a Long Short-Term Memory (LSTM) architecture (Hochreiter & Schmidhuber, 1997). LSTMs are particularly well-suited for partially observable environments because their gated architecture enables them to maintain a compressed representation of observation history. This memory mechanism allows the agent to approximate the true environment state from partial observations, facilitating more informed action selection. Previous work has demonstrated the effectiveness of recurrent architectures in similar settings. Hausknecht & Stone (2017) showed that integrating LSTM into DQN enables learning effective policies with single-frame observations, eliminating the need for the four-frame stacking used in the original DQN. More recently, Ni et al. (2022) demonstrated that recurrent model-free RL serves as a strong baseline across various POMDPs, often matching or outperforming problem-specific approaches. Following the successful application of LSTMs in penetration testing by Zhou et al. (2021), Li et al., and Ren et al. (2024), we evaluate this established approach as a key baseline to include. We use the LSTM-based PPO variant from the `sb3-contrib` repository (Raffin et al., 2021). Going forward, we abbreviate PPO with LSTM as PPO-LSTM.

### 4.2.5 Transformers

The transformer architecture, initially designed for sequence transduction tasks (Vaswani et al., 2017), has shown equal success in a variety of domains such as computer vision (Dosovitskiy et al., 2020) and genomics (Consens et al., 2025). Applying transformers to RL has been challenging, especially due to of the instability they showed initially (Parisotto et al., 2020). Pleines et al. (2025) have successfully applied the Transformer-XL architecture (Dai et al., 2019) to memory tasks in RL. Unlike frame-stacking which provides fixed-window history or LSTMs which compress information through gating, transformers can attend to any part of the observation sequence (Vaswani et al., 2017). While transformers have shown promise in partially observable sequential decision-making tasks, their application to penetration testing remains relatively unexplored compared to LSTM-based approaches. We use the implementation provided in the `cleanRL` (Huang et al., 2022) repository.

### 4.3 Hyperparameter Tuning

Since this study involves a systematic comparison of different algorithms, we conduct a hyperparameter search for each algorithm, to enable a fair comparison. We use Optuna as our optimization framework (Akiba et al., 2019). The hyperparameters are sampled using Optuna's Tree-structured Parzen Estimator (TPE) sampler, a Bayesian optimization method, which efficiently learns from previous trials to suggest promising hyperparameter combinations (Bergstra et al., 2011). Every algorithm has been tuned with a budget of 250 trials. Each trial runs for a maximum of 5 million environment steps, though we employ Optuna's Median Pruner to terminate unpromising trials early after the second evaluation. This pruning mechanism compares a trial's intermediate performance against the median of previous trials around the same evaluation timestep. Importantly, we do not fix the seeds during the hyperparameter search. During the learning phase, the policy is evaluated at intervals of one million steps on freshly generated environments. Each evaluation consists of 100 episodes. The objective function maximizes the mean undiscounted cumulative reward across the 100 evaluation episodes. The full range of hyperparameters tested, along with the best-performing ones for each algorithm, are reported in Appendix A.

## 5 Experiments

### 5.1 Setup

Table 2 summarizes the StochNASim parameters most relevant to this analysis. All environment parameters (e.g., action costs, success probabilities, and sensitive host values) follow the original NASim benchmark (Schwartz & Kurniawatti, 2019) for consistency with prior work; the only modification is varying the network size (5–8 hosts). We now provide justification for these parameter choices and discuss their impact on the learning environment.

**Network Configuration:** We configure networks to vary between 5 and 8 hosts to balance computational tractability with meaningful complexity variation. This range ensures that agents encounter networks of different sizes while maintaining reasonably sized action space and moderate episode lengths. Given the chosen parameters, the size of the action space equals 96 (cf. Section 4.1.4).

**Action Costs and Rewards:** The cost structure in Table 2 reflects realistic penetration testing considerations. Scan actions (cost = 1) represent low-risk reconnaissance activities that are quick to execute but provide limited information. Exploit and privilege escalation actions (cost = 3) are more expensive, reflecting their higher computational overhead, increased detection risk, and potential for causing system disruption. The host value (5) provides modest rewards for gaining access, while sensitive hosts (value = 100) offer substantially higher rewards, creating a clear reward hierarchy that mirrors real-world target prioritization.

**Success Probabilities:** We set exploit and privilege escalation success probabilities to 0.9 rather than making them deterministic. We consider this more realistic as real-world exploits can fail due to factors like timing, system state, or defensive countermeasures. This stochasticity encourages learning more robust policies by exposing agents to occasional failures that must be recognized and retried.

Table 2: Environment parameters for StochNASim used throughout the hyperparameter tuning and experiments.

| Min. Num. Hosts | 5 | Max. Num. Hosts | 8 |
|---|---|---|---|
| Exploit Success Prob. | 0.9 | Priv. Esc. Succcess Prob. | 0.9 |
| Exploit Cost | 3 | Priv. Esc. Cost | 3 |
| Host Value | 5 | Sensitive Host Value | 100 |
| Cost of Scans | 1 | Num. OSes | 2 |
| Num. Services | 2 | Num. Processes | 2 |
| Num. of Sensitive Hosts | 2 | | |

**Operating System and Service Diversity:** We limit the environment to 2 operating systems, 2 services, and 2 processes per category. While this may seem restrictive compared to real-world diversity, it provides sufficient complexity for our comparative study while ensuring that hyperparameter tuning remains computationally feasible. This simplified setup allows us to focus on the core challenge of partial observability without being overwhelmed by combinatorial explosion of possible configurations.

**Step Limit:** The original NASim environment used a step limit of 1000 for networks of size 5 and 8. We chose to increase the step limit to 5000, to avoid making the problem artificially easier by discarding episodes too early (Patterson et al., 2024). The method employed to decide on this number was to run a random agent for 100,000 episodes in the environment. We then took the average number of steps required per episode and multiplied it by an order of magnitude and rounded it up.

## 5.2 Evaluating Hyperparameters

After establishing these environment parameters, we conducted hyperparameter optimization for each algorithm using the methodology established in Section 4.3. Following the hyperparameter search, we select, for each algorithm, the parameters that achieved the highest final evaluation score and assess their performance over a new training run across five control seeds, with a budget of 5 million steps. Using this set of control seeds, we aim to demonstrate the stability of learning and ensure that the selected hyperparameters are not overfit to a single seed. During training, we also evaluate the policy on separate, newly generated, environments at intervals of 500k steps for 100 episodes. We chose such a large amount of evaluation episodes to account for variable episode lengths across network sizes. This regular evaluation serves multiple purposes: it allows us to track training progress over time and validate that our selected hyperparameters lead to consistent improvement, rather than just achieving good final scores by chance. Fig. 3a showcases the training performance of the different algorithms in the form of learning curves. First, we observe that PPO-AO converges twice as fast as PPO-TrXL (1M vs. 2M steps), and four times faster than the remaining methods (4M steps), while achieving a larger cumulative reward. Second, PPO consistently reaches the end of the environment. Although it outperforms a random policy, demonstrating that the agent has acquired some task-relevant behaviour, PPO requires significantly more steps compared to the other methods, thus learning a policy we consider suboptimal. All algorithms underwent identical hyperparameter optimization procedures (250 trials each, detailed in Section 4.3 and Appendix A). Our fANOVA analysis (Figure 8) reveals that for PPO-TrXL and PPO-LSTM, architectural parameters (memory length, positional encoding, hidden size) collectively contribute $<15\%$ of performance variance despite systematic exploration. This suggests the performance differences reflect algorithm-task fit rather than hyperparameter tuning artifacts. It further underscores that solving a POMDP is markedly harder without a mechanism to integrate past observations. Fig. 3b shows the IQM Normalized Score achieved during the intermediary evaluations, plotted using the `rliable` library (Agarwal et al., 2021) for statistically robust performance comparison. Finally, Fig. 3c shows the mean number of steps, during the evaluation periods, required to complete the task of gaining root access on both sensitive hosts. What is notable from this plot is how close PPO-TrXL, PPO-AO and PPO-FS sit together. All three reach the end of an episode in 15-20 steps on average, with cumulative rewards between 140-180. The main takeaway from Fig. 3b and Fig. 3c is that while the top three algorithms reach a solution in roughly the same number of steps, they converge to distinct cumulative rewards, suggesting that their learned policies differ, which warrants a detailed analysis of the learned behaviour.

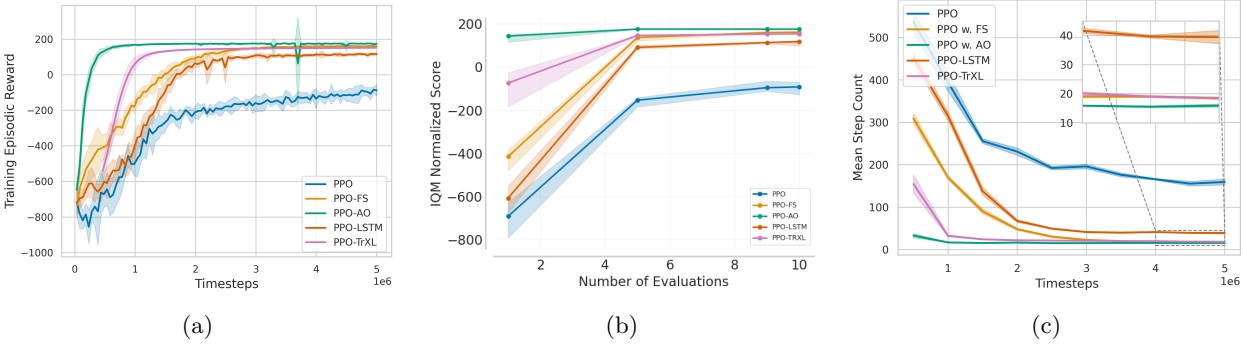

Figure 3: All results are aggregated over 5 seeds. (a) Learning curves of selected algorithms. (b) IQM Normalized Score achieved during intermediary policy evaluations on separate set of environments during training runs. (c) Mean step count to reach the end of a given scenario during intermediary policy evaluations.

## 5.3 Evaluating Learned Policies

While the training curves provide insights into algorithm efficiency and convergence, they do not reveal the qualitative differences in the behaviours the agents learned. To gain further insights into how these policies actually operate we perform additional experiments. To start off, we look at the performance given specific network sizes, with the goal of investigating whether a larger network size has significant influence on the overall achieved reward. To assess performance by network size we load all trained models. Per algorithm we have five models, one for every evaluation seed. We then collect 50 episodes of data per network size. We visualize this data in one box plot per network size, per algorithm, in Fig. 4. This plot gives us better insight into the variance of learned policies. We further confirm that PPO-AO was able to learn the best performing policies. We also observe that every algorithm, except PPO-LSTM, achieves its highest mean reward on networks composed of 7 hosts. We attribute this to the way the networks are generated (Section 4.1.7). With a maximum of 5 hosts per subnet and the DMZ and sensitive hosts occupying separate subnets, the 7th host's placement becomes highly predictable, as it consistently appears as the final host in the user subnet. The addition of an 8th host triggers the creation of an additional subnet, fundamentally altering the network topology. This predictable structure enables agents to develop effective heuristics, such as prioritizing the 7th host when identifying targets for the final two required exploits. Since our evaluation spans four network sizes (5-8 hosts), the 7th host represents the final host in 25% of all scenarios, making it a reliable target. In contrast, networks with 8 hosts introduce additional complexity that degrades performance across all algorithms except vanilla PPO. We attribute this behaviour to the high entropy of the policy, which is required due to not making use of any mechanism to handle the partial observability of the environment. PPO-LSTM experiences the most pronounced decline, showing difficulties to reliably navigate to further subnets.

Beyond performance differences, we also observe qualitative differences in the policies that the different algorithms learn. We showcase this in two ways. First, we analyse the action type distribution across a number of episodes. Second, we take a closer look at an action sequence by investigating the action being taken per timestep. To create the action type distributions, we load the best overall model for each algorithm. This model was determined by evaluating which one achieves the highest mean reward over 100 episodes. The environment is seeded with the same seed for every algorithm which results in the same sequence of initial states when resetting the environment after every episode. This allows for a better and fairer comparison. With these models loaded, we again collect 50 episodes of data, consisting of the executed action, whether it was successful, and the obtained reward. Next, we compute the proportions for each action type. Fig. 5 reveals stark differences in learned policies. The most significant contrast is between PPO-TrXL and the other algorithms. PPO-TrXL appears to have learned no scanning behaviour, instead adopting a brute-force strategy to navigate the network. In contrast, we observe that PPO-AO spends most of its steps, compared to the other algorithms, on scan actions. PPO-FS sits somewhere between PPO-AO and PPO-TrXL, performing only a small amount of scanning, while over half of the actions are executing

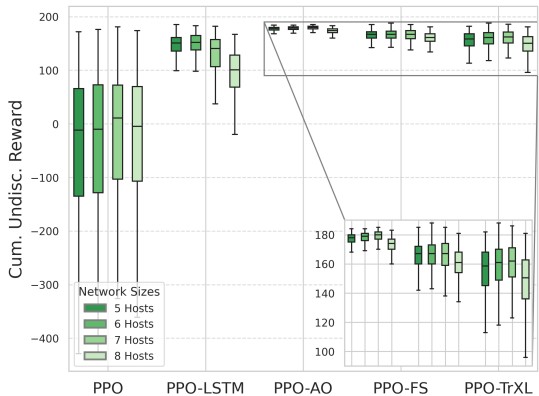

Figure 4: Performance sensitivity analysis across network sizes. Box plots show cumulative undiscounted reward distributions over 50 episodes per network size using the best-performing model from each algorithm. PPO-AO consistently outperforms other methods across all network sizes.

exploit actions. When examining PPO's action distribution, we observe a fairly even spread across all action types. We interpret this as evidence of policy uncertainty stemming from the absence of any history-tracking mechanism. Without access to historical information, the agent cannot leverage prior observations to inform its decision-making, resulting in a highly stochastic policy that lacks the strategic focus demonstrated by methods with memory capabilities.

To explore the learned policies more thoroughly, we select one sequence out of the 50 collected episodes that's representative of the performance of each algorithm, and plot a detailed breakdown of the action sequence in Fig. 6. This confirms the observation from Fig. 3, that while PPO-AO, PPO-FS, and PPO-TrXL achieved similar cumulative results, using a similar number of steps, they learned very different policies.

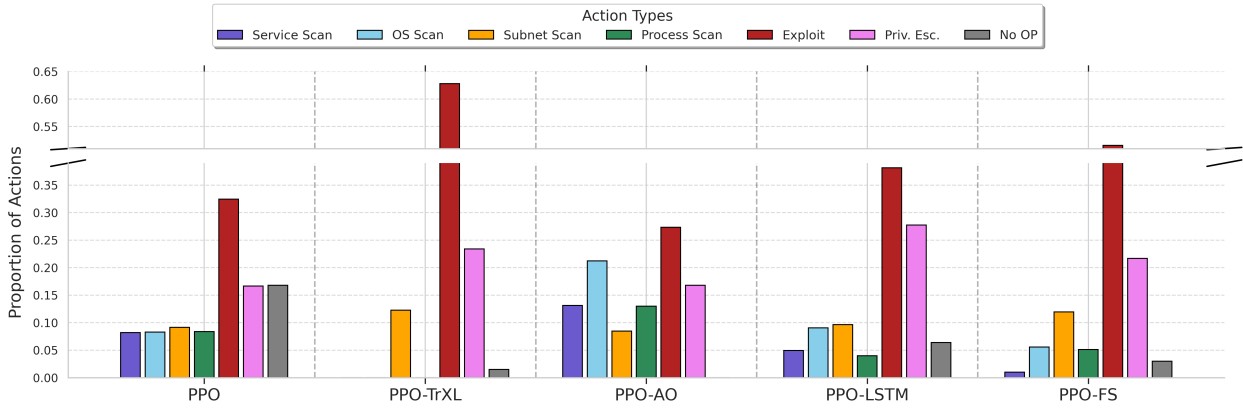

Figure 5: Action type distribution per algorithm, revealing distinct learned strategies despite similar performance outcomes. Data gathered over 50 episodes on separate environments

## 5.4 Stochasticity Analysis

Since NASim only allows learning on fixed-sized networks, we contribute a wrapper to enable fair comparison between environments. This wrapper generates four scenarios at the beginning, one for each network size (5-8), and then cycles through them during training. At each episode, we pick one of these four scenarios; unlike StochNASim, we reuse the same four scenarios rather than generating new ones. We perform five training runs over one million total steps for each environment using PPO-AO, as discussed in Section 4.2.3.

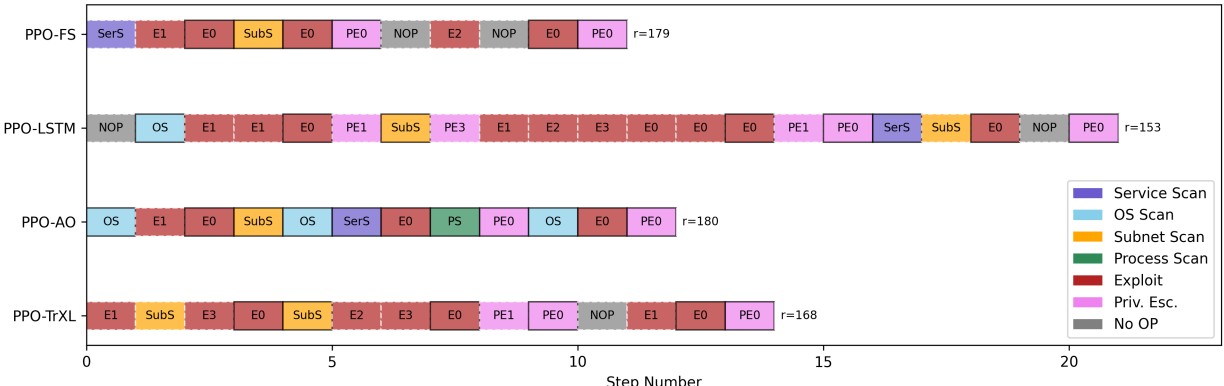

Figure 6: Representative action sequences on a 5-host network. Colours indicate action types, black outlines show success, dotted white show failures. Episode rewards (r=X) and lengths demonstrate strategy efficiency differences.

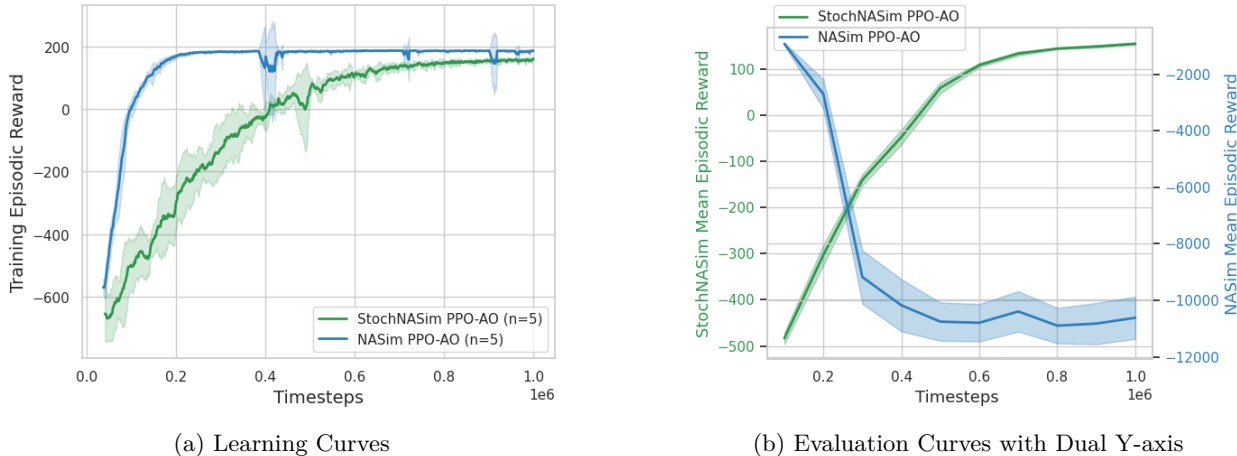

(a) Learning Curves

(b) Evaluation Curves with Dual Y-axis

Figure 7: Training and evaluation performance comparison of PPO-AO trained StochNASim and NASim. Results aggregated over 5 seeds. Policies learned in NASim transfer poorly to unseen network configurations.

During each run, at intervals of 100k steps, the policy is evaluated on a different set of network configurations. The hyperparameters can be found in Appendix A.

This experimental setup reveals striking differences in learning dynamics between the two environments. Figure 7a shows that learning in NASim is simpler: we see less variance and the policy converges after 250k steps. In StochNASim, with an unbounded set of initial states, the policy requires the full training time to converge. To understand why fixed-scenario training leads to overfitting, consider the interaction between limited initial-state diversity and partial observability. With only four initial states, each network has a fixed topology with static host properties. Scanning the same host in the same scenario always reveals the same OS and services. During training, the agent repeatedly observes identical observation sequences. In a POMDP, this enables a particularly problematic form of memorization: rather than learning to aggregate partial observations into belief states, the agent can simply memorize observation-action sequence mappings. This lookup-table strategy succeeds on training scenarios but fails catastrophically on novel configurations where the memorized sequences are invalid. We quantify this train-test generalization gap in Figure 7b: PPO-AO trained on NASim achieves 195±5 mean reward on the four training scenarios but fails catastrophically when evaluated on 100 novel StochNASim-generated networks. In contrast, PPO-AO trained on StochNASim maintains consistent performance: 167±5 on both training and test scenarios. This demonstrates that

stochastic generation forces the agent to learn genuinely generalizable information-aggregation strategies rather than scenario-specific sequence memorization.

## 6 Discussion

The experimental results highlight several important trends and insights. First, the use of StochNASim leads to the ability to generalize over many different generated networks, leading to robust policies that can easily be transferred between configurations, as shown by our experiments in Section 5.4. Second, within the selected algorithms, PPO-AO consistently achieved the highest performance, learning policies that efficiently balance information gathering and exploitation. Despite their architectural complexity, LSTM and TrXL underperformed compared to feedforward PPO variants, with PPO-TrXL notably avoiding scanning altogether. This is an important finding, especially for practical applications, considering the computational overhead that comes with training recurrent- and transformer-based architectures. Further, these findings raise important questions about what constitutes effective penetration testing policies in partially observable environments.

To interpret these results, we must first establish what characterizes effective penetration testing behaviour. An optimal policy should achieve the highest cumulative reward by eliminating the use of unnecessary actions, since all actions incur penalties, with exploit and privilege escalation actions incurring higher costs than scans. The optimal policy thus involves systematically gathering information through scans, retaining this information to avoid redundant scanning, and then selecting the appropriate exploit or privilege escalation action based on the gathered intelligence.

A key finding of our study is that the nature of this task does not require complex nor computationally expensive approaches such as LSTM or TrXL. Given the information-retrieval nature of the environment, simply encoding observation history as augmented inputs proves highly effective. While these conclusions are established within the on-policy, actor-critic framework of PPO—the current standard for this domain—they demonstrate that architectural complexity does not inherently guarantee better reasoning over discovered facts.

Interestingly, our manual inspection of learned policies reveals distinct behavioural patterns among the policies learnt by the different algorithms we studied. PPO-TrXL learned a brute-force policy, sequentially trying every exploit until gaining user access, then attempting every privilege escalation action until achieving root access. PPO-LSTM does make use of scans, but overall, the LSTM architecture was not able to learn a good representation of the history. Requiring almost twice the amount of steps to exploit all the sensitive hosts. PPO-FS demonstrated more refined behaviour, learning to match privilege escalation actions to discovered information but still executing exploits sequentially. Only PPO-AO achieved what we consider an optimal policy, executing the task efficiently with minimal redundant actions.

These results highlight a crucial insight: not tracking the observation history leads to significantly worse policies that would be impractical in real-world environments. However, the solution need not be computationally expensive: simple observation augmentation outperforms sophisticated memory architectures for this particular domain.

Looking towards future work, a key limitation of our current approach becomes apparent in more realistic settings, such as dynamic network environments where host states can change during an episode. These changes may include hosts shutting down, firewall rules being updated, or paths becoming unavailable, which can render previously collected observations invalid. In such cases, hand-crafted observation augmentations might become brittle and infeasible to maintain. Moreover, in many real-world scenarios, it is impractical or intractable to design augmentations manually. Ideally, we would rely on the agent itself to learn effective state representations that integrate relevant historical information and adapt to evolving conditions. Architectures such as LSTMs and TrXL are designed to support such functionality by learning from sequences directly, but our empirical results show that their performance remains limited in this setting and context. This suggests that these models may be struggling to capture the specific type of memory, reasoning, or structural understanding required for effective decision-making in partially observable and dynamic environments. An interesting avenue for future work lies in developing or adapting history-aware models, with stronger inductive

biases or explicit memory mechanisms, that can better handle such evolving, partially observed domains. This is especially interesting in conjunction with larger networks (>20 hosts), which might also reveal whether the observed architectural rankings generalize beyond the current scale. Furthermore, investigating whether the superiority of simple history aggregation holds for off-policy algorithms like DQN or SAC remains an open question, as these methods may interact differently with the temporal consistency of augmented state representations.

## 7    Conclusion

In this work, we modelled penetration testing as a partially observable sequential decision-making problem in stochastic environments encompassing networks of varying sizes. To address the limitations of existing simulators that use fixed network configurations, we developed StochNASim, a stochastic extension of NASim that generates new network topologies with varying host properties and network sizes for each episode. Using this environment, we systematically compared different approaches for addressing partial observability, ranging from no memory mechanism (baseline) to augmented observations, frame-stacking, LSTM and TrXL architectures. Our findings reveal that this task is well-suited for simpler memory mechanisms that track observation history through direct aggregation. Interestingly, complex architectures like LSTM and TrXL failed to learn sophisticated policies, instead developing undesirable brute-force approaches that sequentially attempt all possible actions. Notably, PPO-AO outperformed all other methods, learning the most efficient and human-like penetration testing strategies while converging up to four times faster than competing approaches. The stochastic nature of StochNASim proved crucial for developing robust policies. Our comparison between fixed and stochastic environments demonstrated that policies trained on static configurations show poor generalization to novel scenarios, while those trained in our variable environment maintain consistent performance across different network configurations. This highlights the importance of stochastic training environments for learning policies applicable to real-world penetration testing scenarios. Crucially, our results demonstrate the importance of looking beyond learning curves and final rewards when evaluating learned policies. We found that algorithms achieving similar quantitative performance can learn vastly different behavioural strategies, highlighting the value of qualitative policy analysis in understanding algorithmic effectiveness. While our experiments were conducted in simulation using on-policy methods, these insights are particularly relevant as the field moves toward real-world penetration testing applications, where efficient, interpretable, reliable, and robust policies are essential for practical deployment.

### Acknowledgment

This research was funded by the Royal Higher Institute of Defence under the project DAP23/05. This work was supported by the Flemish Government under the "Onderzoeksprogramma Artificiële Intelligentie (AI) Vlaanderen" program. The resources and services used in this work were, in part, provided by the VSC (Flemish Supercomputer Center), funded by the Research Foundation - Flanders (FWO) and the Flemish Government. Pieter Libin acknowledges support from the Research council of the Vrije Universiteit Brussel (OZR-VUB) via grant number OZR3863BOF. We also thank Florent Delgrange and Raphael Avalos for their invaluable insights, Mehrdad Asadi for feedback on visualizations and Arbia Riahi for proof reading, as well as anonymous reviewers for their useful feedback.

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

Table 3: Hyperparameter search ranges for PPO, PPO-FS and PPO-AO. Best performing parameter values for PPO are marked in **bold**, underlined for PPO-FS and in *italics* for PPO-AO.

| Hyperparameter | Values |
|---|---|
| Batch Size | 64, *128*, **256**, 512 |
| Number of Steps | 128, 256, 512, ***1024***, 2048 |
| Discount Factor ($\gamma$) | 0.95, **0.97**, 0.99, 0,995, *0.999* |
| Learning Rate | 3e-5, 1e-4, ***3e-4***, 1e-3, 3e-3 |
| Entropy Coefficient | 1e-3, 5e-3, *1e-2*, 5e-2, **1e-1** |
| Clip Range | 0.1, *0.2*, 0.3, **0.4** |
| Number of Epochs | **5**, *10*, 20 |
| GAE Lambda | ***0.9***, 0.95, 0.99 |
| Maximum Gradient Norm | 0.5, **0.6**, 0.7, 0.8, 0.9, *1*, 2 |
| Value Function Coefficient | ***0.3***, 0.5, 0.7 |
| *Network Parameters* | |
| Activation Function | **tanh**, *ReLU* |
| Orthogonal Initialization | False (fixed) |
| Network Architecture | Tiny: $\pi$=[64], vf=[64] |
| | Small: $\pi$=[64, 64], vf=[64, 64] |
| | Medium: $\pi$=[128, 128], vf=[128, 128] |
| | **Large**: $\pi$=[256], vf=[256] |
| | *Very large*: $\pi$=[256, 256], vf=[256, 256] |
| *PPO-FS specific parameters* | |
| Frame Stack | 4, 8, 16, 32 |

## A Hyperparameters

For PPO-TrXL, we've taken most of the hyperparameter ranges from the original implementation as they are described the the work of Pleines et al. (2025). Some modification have been made to the number of steps performed per rollout and the number of environments use. We used 768 steps per rollout and 8 environments. This choice stems from memory restrictions on the GPUs that we used. PPO-TrXL was trained on a cluster of four NVIDIA A100 and H100 respectively. When testing the implementation, we found that not collecting enough steps from the environment may result in unfinished episodes, which will terminate the training because there is no data to bootstrap on. The remaining algorithms have been trained on a clusters composed of Intel Xeon Gold 6148 (Skylake). Tables 3–5 showcase the hyperparameter ranges that were used during the search initially described in Section 4.3. To tune the hyperparameters of PPO, PPO-AO, PPO-FS and PPO-LSTM, we use the `rl-baselines3-zoo` (Raffin, 2020) library. For PPO-TrXL, we wrote a hyperparameter-tuning script adhering to the same structure.

### A.1 Sensitivity Analysis

Fig. 8 showcases the importances of the different hyperparameters for each algorithm established in Section 4.2. We can see that for PPO without any history aggregation or reconstruction, the most important parameter is the entropy coefficient. It controls to which degree the policy is going to pick random actions, instead of the best one. The entropy accounts for half of the total variability of the trials. This reflects an inherent uncertainty that is present, since the decision which action to take solely relies on the last observation. When comparing the importances for PPO to frame-stacking and the augmented observations, we can see that the entropy coefficient matters a lot less. This is due to more context provided in the observations, which helps creating a representation and therefore requiring less randomness from policy. They can better rely on the observations they receive. Something else that is worth highlighting is the importance of the number of stacked frames (or rather observations in our case) for PPO-FS. We would assume that the frame-stack parameter $f_n$ played a more important role, but it accounts for less than 5%

Table 4: Hyperparameter search ranges for PPO-LSTM optimization. Best performing parameter values are marked in **bold**.

| Hyperparameter | Values |
| --- | --- |
| Batch Size | 64, 128, 256, **512** |
| Number of Steps | 128, 256, **512**, 1024, 2048 |
| Discount Factor ($\gamma$) | **0.99**, 0,995, 0.999 |
| Learning Rate | 1e-5, 3e-5, 1e-4, 3e-4, **1e-3** |
| Entropy Coefficient | 1e-3, 5e-3, 1e-2, **5e-2**, 1e-1 |
| Clip Range | **0.1**, 0.2, 0.3, 0.4 |
| Number of Epochs | **5**, 10, 20 |
| GAE Lambda | 0.9, **0.95**, 0.98 |
| Maximum Gradient Norm | 0.5, 0.6, 0.7, 0.8, **0.9**, 1, 2 |
| Value Function Coefficient | **0.3**, 0.5, 0.7 |
| *Network Parameters* | |
| Activation Function | **tanh**, ReLU |
| Orthogonal Initialization | False (fixed) |
| LSTM Hidden Size | 64, **128**, 256 |
| Enable Critic LSTM | True, **False** |
| Network Architecture | Small: $\pi$=[64, 64], vf=[64, 64] |
| | Medium: $\pi$=[128, 128], vf=[128, 128] |
| | **Large**: $\pi$=[256], vf=[256] |

Table 5: Hyperparameter search ranges for PPO-TrXL. Best performing parameter values are marked in **bold**.

| Hyperparameter | Values |
| --- | --- |
| Number Mini Batches | 2, 3, **4** |
| Update Epochs | 2, 3, **4** |
| Discount Factor ($\gamma$) | 0.95 0.99, **0,995**, 0.999 |
| Learning Rate init. | **2e-4**, 2.75e-4, 3e-4, 3.5e-4 |
| Learning Rate final (fixed) | 1e-5 |
| Entropy Coef. init. | **1e-4**, 1e-3, 1e-2 |
| Entropy Coef. final (fixed) | 1e-6 |
| Anneal Steps (fixed) | 4020000 |
| Clip Range | **0.1**, 0.2, 0.3 |
| Normalize Advantage | True, **False** |
| GAE Lambda ($\lambda$) | 0.9, **0.95**, 0.99 |
| Maximum Gradient Norm | 0.25, 0.35, **0.5**, 1 |
| Value Function Coefficient | 0.2, **0.3**, 0.5 |
| *TrXL Parameters* | |
| TrXL Num. Layers | 2, 3, **4** |
| TrXL Num. Heads | **1**, 4, 8 |
| TrXL Dimension | 128, **256**, 384 |
| TrXL Memory Length | 128, 256, **512** |
| Positional Encoding | **none**, absolute, learned |

of the total variability. We interpret it that already a small amount of stacked observations are beneficial to enhance the overall performance. Interestingly, for PPO-TrXL, none of the TrXL architecture specific parameters appear to be the most important ones. Concerning PPO-LSTM, the batch size appears to be the most important parameter, accounting for approximately 35% of the total variability. This finding aligns with the unique challenges of training recurrent networks in reinforcement learning settings. Unlike the other PPO variants, LSTM networks require careful management of sequential dependencies and hidden state propagation across time steps. The batch size directly influences how many independent sequences the LSTM processes simultaneously, which is crucial for stable gradient estimation and proper learning of temporal patterns.

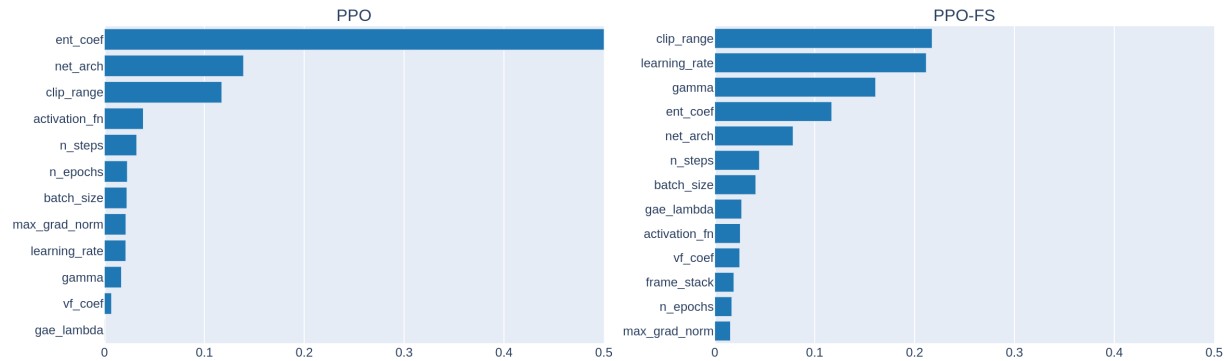

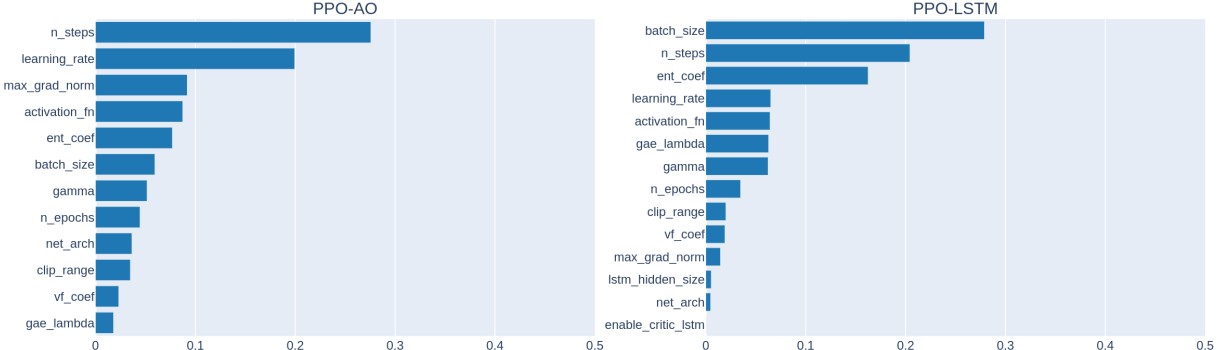

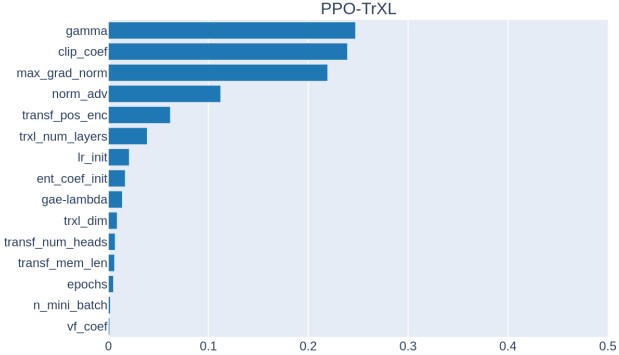

Figure 8: Hyperparameter importance analysis using fANOVA framework (Hutter et al., 2014). Values show the fraction of performance variance explained by each parameter across 250 optimization trials per algorithm.

## B  Additional Notes on Experiments

The seeds that have been used in Section 5.2 are the following: 8258, 710, 6930, 8829, 7602. We simply pass them as an argument to the scripts from `cleanRL` and `stable-baselines3`. The specific version we used for the `stable-baselines3` framework is 2.4. This holds for both the algorithm implementations as well as their hyperparameter tuning framework in `rl-baselines3-zoo`. The seed we employed for the environments regarding the experiments conducted in Section 5.3 is 2. We provide the code for the StochNASim environment and PPO-TrXL in the supplementary material.

