# OpenReview forum: "Learning Robust Penetration Testing Policies under Partial Observability: A systematic evaluation"
_TMLR — Accepted by TMLR_

### Review · Reviewer_GpeA · 2026-01-03

**Summary Of Contributions:**

The paper introduces StochNASim, an extension of NASim that adds (1) stochastic network generation per episode, (2) variable network size (5–8 hosts), and (3) a strict POMDP observation model. Using this environment, the authors systematically evaluate mechanisms for handling partial observability in penetration testing: PPO, PPO-FS, PPO-AO, PPO-LSTM, and PPO-TrXL. PPO-AO emerges as the strongest baseline, achieving significantly faster convergence and more effective behaviour. The paper also provides qualitative analysis of action sequences and demonstrates that stochastic training prevents severe overfitting seen in NASim.

**Additional Comments:**

Overall, this is a strong empirical study with a meaningful new environment and valuable qualitative insights. The main revisions needed are conceptual clarifications (overfitting, observation model), expanded justification of environment parameters, and more careful boundary-setting of claims about algorithm generality. With these addressed, the paper would be a solid TMLR contribution.

**Audience:**

Yes

**Audience Explanation:**

The work is related to the domain of (1) penetration testing and (2) reinforcement learning under partial observability, both of which are related to TMLR's topic on machine learning and control theory. The empirical results and analysis provide potential insights for researchers in these domains to further expand and improve algorithms like PPO to make they more versatile and applicable in real-world scenarios.

**Broader Impact Concerns:**

No ethical implications.

**Claims And Evidence:**

Yes

**Claims Explanation:**

- Well-motivated environment contribution: The paper clearly articulates why fixed-scenario MDP formulations in prior work are unrealistic for penetration testing and demonstrates (Fig. 7, p. 15) that stochastic generation materially improves policy robustness.
- Systematic algorithmic comparison: Hyperparameter tuning with Optuna (250 trials each) and evaluation with multiple seeds is unusually thorough for the penetration-testing RL literature. The fANOVA analysis (Fig. 8, p. 23) adds transparency.
- Clear qualitative policy analysis: Action distributions (Fig. 5, p. 13) and step-wise sequences (Fig. 6, p. 14) reveal strong differences between learned strategies that would not be apparent from returns alone.
- Strong empirical narrative: The contrast between PPO-AO, PPO-FS, and PPO-TrXL shows how history aggregation dominates architectural sophistication for this task.

**Requested Changes:**

Overfitting motivation needs clearer grounding While NASim vs StochNASim results highlight generalization issues, the manuscript does not explicitly analyze why fixed-scenario training overfits nor how this relates to partial observability. A more explicit treatment would clarify the central claim. If possible, please add a small section or paragraph quantifying train–test generalization under NASim and explicitly discuss how limited initial-state diversity inflates memorization.
Observation model justification is incomplete Section 4.1.5 asserts that “observations only contain the direct result of each action” without citing penetration-testing conventions or operational constraints. It also leaves unclear how unknown vs. absent attributes are represented under the element-wise max augmentation in PPO-AO. Please provide explicit justification, ideally referencing pentesting workflows/tools, and clarify:
How negative information is encoded (if at all).
Whether the agent can distinguish “not yet scanned” vs “absent”.
Parameter choices in Table 1 (p. 11) feel arbitrary Values such as success probability = 0.9, cost structure, and the 5–8 host range are qualitatively motivated but not empirically validated.
Algorithmic scope is narrow (all PPO-based) Given the paper’s claim that simple history aggregation outperforms advanced memory architectures, restricting evaluation to PPO variants limits generality. Ideally to show significance of the work, we need to have either one additional baseline (e.g., off-policy methods), or clearly temper the claim to state that conclusions are limited to PPO-style on-policy methods.

---

> ### Author Response · Authors · 2026-02-13
> **Reply to reviewer GpeA**
>
> We thank the reviewer for their thorough and constructive feedback. We are pleased that they found our contributions valuable and our experimental evidence convincing. We have addressed all requested changes as detailed below.
>
> 1) Overfitting Motivation Needs Clearer Grounding
>
> Thank you for pointing this out. We have revised Section 5.4 to provide: (1) explanation of how partial observability enables observation-sequence memorization with fixed scenarios, and (2) added quantitative train-test generalization metrics between the PPO-AO policies trained on NASim and StochNASim.
>
> 2) Observation Model Justification is Incomplete
>
> We agree that Section 4.1.5 could benefit from more clarity. As such, it now includes: (1) citation to penetration testing tools (Nmap) justifying action-specific observations, and (2) explicit encoding scheme clarifying that observations are transient and that unscanned/absent attributes are both encoded as zeros (agent cannot distinguish them).
>
> 3) Parameter Choices Feel Arbitrary
>
> We appreciate this observation. However, we would like to clarify that our parameter choices are not arbitrary but are deliberately adapted from the original NASim benchmark (Schwartz & Kurniawatti, 2019) to maintain consistency and comparability with prior work in this domain. We have highlighted this better in the revised manuscript.
>
> 4) Algorithmic Scope is Narrow (All PPO-based)
>
> We appreciate the reviewer's point regarding the algorithmic scope. We chose to focus exclusively on PPO-based variants because PPO is currently the de facto standard for automated penetration testing due to its stability and performance in complex strategic environments. By keeping the base algorithm constant, we were able to conduct a rigorous, high-budget hyperparameter search (250 trials per variant) specifically focused on the interaction between partial observability and memory. However, we agree that the findings are most strongly validated within the context of on-policy, actor-critic frameworks. We have updated our Introduction, Discussion and Conclusion sections to temper our claims, specifying that while simple history aggregation is highly effective for PPO-style agents in this domain, further study is required to confirm if these insights transfer to off-policy architectures.

---

### Review · Reviewer_XoRx · 2026-01-29

**Summary Of Contributions:**

The paper focuses on automated penetration testing by formalizing network penetration testing as a Partially Observable Markov Decision Process (POMDP), moving beyond fully observable MDP formulations. The primary technical contribution is StochNASim, an adapted evaluation environment that introduces partial observability and stochasticity for building more representative and closer to real-world benchmarks. The second contribution is the systematic empirical analysis over a number of Reinforcement Learning algorithms as the penetration test agent in handling the partial observability. showing that PPO-AO (PPO with augmented observations) outperformed other methods.

**Audience:**

Yes

**Audience Explanation:**

The new simulation environment StochNASim built on top of existing NASim framework with improvement targeting at addressing the practical limitation of the existing framework, lack of partial observability and stochasticity, are worth noting by the community. The empirical experiments are comprehensive, which also worth the attention from researchers interested in this topic.

**Claims And Evidence:**

Yes

**Claims Explanation:**

As a empirical work, the conclusion PPO-AO outperforms other methods are well supported by experiments designed to showcase (1) reward convergence speed during training (2) sensitivity analysis and (3) stochasticity analysis, with results presented by aggregating 5 runs with different controlled seed.

**Requested Changes:**

The writing of this paper is a little hard to follow, particularly for readers that are not familiar with the topic. The most notable issue is that it is hard for the reader to build a connection of the main problem considered in this paper and the related works mentioned.

For example, by reading section 2 alone, it is hard to get the connection how MDP, partial observability, and RL are related to Penetration Testing. Without sufficient knowledge background, seeing "The sequential decision-making nature of penetration testing makes it particularly suited for (its) automation by RL" is not self-evident. One suggestion is to reverse the order of presentation: First, introduce what is penetration testing. Next, explain what are the key components and steps for penetration testing. Then explain the technical side of MDP, partial observability, and RL, helping the readers to build the connection of what role each of these technical component can play in penetration testing. E.g., what part in penetration testing is modeled by MDP? what component are partially observable? what purpose is RL used for? The authors should present these ideas clearly to the audience.

Related works currently seems like a marketplace for jargons. This may partially due to the issue of section 2, where audience may not get a full picture of the problem considered in this paper. Nonetheless, the authors may take a step back in writing and thinking about a smoother logic flow to connect the paragraphs presented in section 3. For example, there are many problems come out of my mind when reading this section alone. E.g.,
1. Full observability & partial observability, offensive & defensive agents, lateral movement and privilege escalation v.s. full penetration testing ==> what are the differences?
2. "While these environments provide partial observability, most .. opted for the simpler fully observable formulation" ==> why? what are the consideration in these works? It sounds odd to call it "justifiable" for "hard to achieve good performance under partial observability".
3. The following logic order is also odd. (i) This trend left a significant gap .... ==> (ii) we systematically compare .. ===> (ii) Several recent works have begun to address partial observability ... Wouldn't (i) ==> (iii) ==> (ii) make more sense? Otherwise, what's the additive value of your work compared to (iii)?

In methodology section, I suggest the authors to include a comprehensive illustration of the base simulator (NASim) than what the paper currently offered before talking about the contribution (StochNASim). Since StochNASim is built upon NASim, this can help emphasize and highlight the difference. After all, the audience will appreciate StochNASim more if they have a better idea on what NASim is when they are presented with StochNASim.

---

> ### Author Response · Authors · 2026-02-13
> **Reply to reviewer XoRx**
>
> We thank the reviewer for their thorough and constructive feedback. We are pleased that they found our contributions valuable and our experimental evidence convincing. The reviewer's suggestions have significantly improved the paper's accessibility while maintaining technical rigor.
>
> 1) Restructuring Section 2 (Background)
>
> We have restructured Section 2 following the reviewer's suggestion. The section now: (1) begins with penetration testing (Section 2.1), introducing the domain problem before mathematical formalism; (2) establishes the explicit mapping from pentesting concepts to RL components (states, actions, observations, rewards, transitions) at the end of Section 2.1; and (3) introduces the formal frameworks (MDPs, POMDPs, RL) with clear motivation for their relevance. This reordering should help readers "build a connection of what role each of these technical components can play in penetration testing."
>
> 2) Improving Section 3 (Related Work)
>
> We have revised Section 3 to address all three concerns:
>
> 1. Clarifying terminology: Added explicit definitions of offensive vs. defensive agents and the distinction between lateral movement/privilege escalation vs. full penetration testing workflow where necessary.
> 2. Removing unsupported speculation: Removed "justifiable" and all speculation about why prior work chose full observability. We now simply state factually what was done: "most subsequent research has used fully observable MDP formulations" followed by "In contrast, we focus on POMDPs..."
> 3. Improving logical flow: Reordered to: gap identified → recent work addressing it → limitations of that work → our contribution. This structure should make it clearer how our work builds upon and extends beyond recent efforts.
>
> 4) Enhancing Section 4 (Methodology)
>
> We have added a comparison table contrasting NASim's original capabilities with StochNASim's extensions, providing the "comprehensive illustration of the base simulator" that was requested.

---

### Review · Reviewer_T5Pt · 2026-01-31

**Summary Of Contributions:**

The paper formulates automated penetration testing as partially observable RL in a network-simulation environment where key host properties are discovered over time and remain static within an episode. It introduces/uses a stochastic, variable-size network benchmark and compares PPO with different memory mechanisms: frame stacking, explicit belief-style aggregation (union of discovered facts), LSTM, and a Transformer-XL (TrXL) variant. Experiments (on small networks, ~5 to 8 hosts) suggest that the explicit fact-accumulation memory yields the best performance and faster convergence, while LSTM/TrXL underperform in their implementation leading the authors to argue that handcrafted symbolic-style memory is particularly effective for this domain and setup.

**Audience:**

Yes

**Audience Explanation:**

Yes it is of interest to the community.

**Claims And Evidence:**

No

**Claims Explanation:**

Disclaimer: I am not a domain expert in cyber-security. These are my thoughts however after reading the paper.

Not fully. The evidence seems partially supportive within the authors’ small, specific setup, but not yet accurate/convincing enough to justify the broader framing and stronger conclusions.

Here are my thoughts on why this is so.

1. Convincing evidence is limited by scale and realistic benchmarking : 5 - 8 host networks feel too small to confidently claim general usefulness or robustness. That makes “works well without foundation models” vulnerable to being a small-environment artifact.

2. Claims vs evidence mismatch for Transformers: The experiments show TrXL/LSTM underperform in their setting, but without careful ablations (encoding/tokenization, memory length, normalization, stability tricks), it’s not convincing to generalize to “Transformers are bad for this domain.”

3. The environment’s “static-within-episode” properties + a handcrafted union-of-facts memory makes PPO-AO’s advantage plausible, but also means the comparison may be biased toward that inductive bias unless the paper disentangles “explicit belief cache” vs “learned memory.”

4. Inconsistent speedup wording (“three times” vs “twice” vs “four times”) weakens the “clear evidence” part.

**Requested Changes:**

These are a few suggested changes that could address the claims part.

1. Evaluate on larger networks (more hosts/services) and/or more diverse scenarios. A distribution shift study (e.g., training on one set of network configurations and testing on different topologies, OS/service mixes, or vulnerability distributions) would substantially strengthen any claims of robustness and general applicability.

2. While the paper appropriately references known instability issues for Transformers in RL, it should be more cautious in its phrasing: the results demonstrate that TrXL underperformed in this particular setting and implementation, rather than supporting a broad conclusion that “Transformers are bad for this domain.” My suggested fix is either, (i) temper the claims in the abstract/conclusion, or (ii) add targeted ablations / implementation details to isolate whether the underperformance is due to model class vs. training/representation choices.

3. In this environment, many properties are static within an episode (e.g., OS/services/processes), so partial observability largely reduces to accumulating discovered facts (a belief-state cache). PPO-AO effectively implements an explicit union-of-facts memory, so its strong performance may reflect a favorable inductive bias, while LSTM/TrXL may struggle to learn the same symbolic accumulation from raw inputs. Moreover, the padded $(m+1) \times n$ observation matrix could create representation mismatch: whether it’s flattened, tokenized, or embedded can strongly affect sequence models. As a suggested fix add ablations on (a) static vs. dynamic information, (b) TrXL/LSTM encoding choices, and (c) memory length / alternative encodings to test whether conclusions hold.

Pls note that the above fixes/changes are recommended. However, please feel free to demonstrate the truthfulness of the claims with other easily implementable fixes if there are any.

---

> ### Author Response · Authors · 2026-02-13
> **Reply 1/2 to reviewer T5Pt**
>
> We thank reviewer T5Pt for the insightful feedback that has helped us better articulate the scope and contribution of our work. We address all of the remarks, numbered by the paragraph in which the reviewer structured their comments.
>
> 1) Scale and benchmarking realism
>
> We thank the reviewer for this suggestion. However, we believe there may be a misunderstanding regarding the scope and positioning of our contribution. Our paper does not claim real-world deployment readiness or general applicability across arbitrary network configurations. Rather, our goal is to take the next step in advancing the state of the art for RL-based penetration testing by introducing and systematically evaluating approaches to handle partial observability. This remains a challenge that much of the prior literature either ignores entirely or addresses only superficially (e.g., Terranova et al. acknowledge POMDPs but revert to MDP-based algorithms; many NASim-based studies operate in fully observable mode).
>
> Regarding network scale: our scenarios (5–8 hosts) are consistent with the benchmarks commonly used in this community. Moreover, from a partially observable perspective, our scenarios go beyond the state of the art in this regard. The original NASim benchmarks (Schwartz & Kurniawatti, 2019), NASimEmu (Janisch et al., 2023), and recent evaluation studies (e.g., arXiv:2407.15656) all operate at comparable scales. Critically, no prior work in this space has conducted a systematic comparison of POMDP-handling architectures (frame-stacking, augmented observations, recurrent, and transformer-based) at any scale. Our primary contribution is this architectural comparison under partial observability, and while we recognise that scalability is an important consideration, we believe the insights from this comparison are valuable in their own right and provide a necessary foundation for future work at larger scales.
>
> Moreover, while the host count is modest, the resulting RL problem is far from trivial. Our stochastic scenarios feature a 96-dimensional action space, comparable to or exceeding many standard POMDP benchmarks used in the broader RL literature. The stochastic action outcomes (Bernoulli success probabilities on exploits and privilege escalations) generate a large space of possible trajectories, meaning the agent cannot rely on memorised action sequences (cf. Section 5.4). Combined with partial observability, where the agent must learn active information-gathering strategies before it can act effectively, these scenarios present a genuinely challenging decision-making problem, even at the current network scale.
>
> We stress that our contribution is methodological: comparing memory mechanisms for partially observable penetration testing. For this algorithmic comparison, our scale enables rigorous hyperparameter tuning (250 trials × 5 algorithms = 1,250 training runs), setting our work apart from prior studies while remaining computationally tractable. A less thorough comparison on larger networks would arguably yield less reliable insights than the thorough analysis we present at the current scale.
>
> Regarding the remarks on distribution shift, we argue that StochNASim already incorporates a meaningful form of generalization testing by design. Both training and evaluation use stochastically generated networks with different random seeds, meaning that every evaluation episode tests the agent on an unseen network configuration with different topologies, host properties, and service distributions. In other words, the agent is never evaluated on configurations it has seen during training. Figure 7b demonstrates this concretely: policies trained in StochNASim maintain their performance on held-out evaluation environments, whereas policies trained on NASim's fixed scenarios show severe degradation when tested on novel configurations. This directly evidences the robustness benefit of stochastic training and constitutes a meaningful form of distribution shift evaluation. We acknowledge that more structured transfer studies (e.g., training on small networks, testing on larger ones) would provide further insights, and we now discuss this as a natural next step in the revised paper.
>
> We do acknowledge that understanding whether the observed architectural rankings hold at larger scale is a legitimate and important question. We will revise the paper to: (a) make the scope of our claims more explicit in the abstract and conclusion, avoiding language that could be read as implying broader generalizability than we demonstrate; and (b) expand the Discussion section to position larger-scale evaluation and distribution shift studies as concrete future work, noting that StochNASim's variable-size observation space (the padded (m+1)×n design) was engineered with precisely this extensibility in mind.

---

> ### Author Response · Authors · 2026-02-13
> **Reply 2/2 to reviewer T5Pt**
>
> 2) Transformer Performance
>
> We agree with the reviewer that our phrasing regarding Transformer performance should be more measured, and we will revise the abstract and conclusion accordingly to make clear that our results reflect TrXL's performance within our specific setting and hyperparameter regime, rather than a blanket conclusion about Transformers in this domain.
>
> That said, we want to emphasise that, in our setting, TrXL was not disadvantaged by a lack of tuning effort. Each algorithm  underwent 250 hyperparameter trials, for a total of 1,250 training runs across all methods. This systematic search covered learning rates, network sizes, and architecture-specific parameters such as memory length and number of attention heads. The underperformance of TrXL was consistent across this broad sweep, which suggests it is not simply an artefact of a single unfortunate configuration.
>
> In the revised paper, we will temper the relevant claims and frame our TrXL results as evidence within this particular experimental context, while noting the alignment with broader theoretical concerns about Transformers in partially observable settings.
>
> 3) Static Properties and Inductive Bias
>
> The reviewer correctly identifies that the static-within-episode properties of our environment create a favorable inductive bias for PPO-AO's explicit memory. We appreciate this observation and have used it to sharpen our contribution in the revised manuscript.
>
> Importantly, this structure reflects realistic penetration testing constraints: host configurations (OS, services, processes) typically remain stable during an assessment. Moreover, as clarified in the revised Section 4.1.5, observations in StochNASim are action-specific and transient. This means that negative results (absent services) are indistinguishable from unscanned attributes, as both are represented as zeros. This is not an artefact of our simulator design but mirrors a real challenge: professional penetration testers maintain persistent notes precisely because transient tool outputs and implicit negatives must be manually aggregated to resolve state ambiguity. PPO-AO's element-wise maximum aggregation mirrors this practice, serving as an effective forgetting prevention mechanism well-suited to this specific signal structure.
>
> In the revised manuscript, we have reframed our contribution to reflect this insight. Rather than claiming universal superiority over learned memory, we now characterise our finding more precisely: for partially observable tasks characterised by transient signals, state aliasing, and monotonic information accumulation, explicit memory mechanisms provide a superior inductive bias over learned memory. While TrXL and LSTMs possess greater architectural capacity, they struggled to learn the specific reasoning required to distinguish "lack of knowledge" from "known absence" from raw observation sequences alone, in our particular setting.
>
> We acknowledge that the targeted ablations the reviewer suggests (e.g. on dynamic vs. static information and alternative encoding choices) are valuable directions. Our revised Discussion (Section 6) now explicitly notes that dynamic scenarios, where host states change mid-episode, would likely favour the adaptive nature of learned memory. However, we believe that identifying a well-characterised class of problems where simple explicit memory suffices is itself a useful contribution, as it allows practitioners to achieve effective strategies without the computational overhead of training complex sequence models.

---

### Decision · Action_Editor_gzfg · 2026-04-10

**Recommendation:** Accept as is

**Audience:**

Yes

**Audience Explanation:**

Even though the paper does not provide novel theoretical insights, it tackles a real-world application of reinforcement learning involving partial observability, which can be of interest to RL practitioners. The problem itself is also interesting from an application perspective, as it deals with security vulnerabilities and cyberattacks in network environments.
That said, the application domain is quite niche, so the targeted TMLR audience should be rather limited.

**Claims And Evidence:**

Yes

**Claims Explanation:**

Overall, I feel that this is a well-written paper. The penetration testing scope is rather limited, but the paper still manages to offer some valuable insights for the problem of penetration testing. In particular, the authors
- motivate the problem of penetration testing.
- introduce StochNASim, an adapted evaluation environment that introduces partial observability and stochasticity for building more representative and closer to real-world benchmarks.
- explain how partial observability affects this problem and why their penetration testing constraints are realistic.
- conduct a systematic empirical analysis over a number of Reinforcement Learning algorithms as the penetration test agent in handling the partial observability, showing the superiority of simple memory augmentation
- show how simple memory augmentation techniques can outperform more complex algorithms, and how it is qualitatively different in terms of action distribution.

All Reviewers agree that their concerns were addressed and favour this paper for acceptance:
- The scale of the experiments is small, but the authors clarified their goal is not real-world deployment readiness but rather advancing the state-of-the-art for penetration testing.
- The authors toned down some of their claims regarding TrXL/LSTM underperformance and PPO-AO’s advantage.
- The authors improved terminology, logical flow, and consistency. They also clarified their methodology.
- The authors clarified points on overfitting motivation and observation model justification.

Experiments of larger scales would have been very interesting, but the authors explained their network sizes are consistent with the benchmarks commonly used in this community. Moreover, from a partially observable perspective, their scenarios go beyond the state of the art in this regard.

For all the above reasons, and given the thorough revision which addressed all concerns raised by the Reviewers, I believe the paper is qualified for acceptance as is.